# Treatment success for patients with tuberculosis receiving care in areas severely affected by Hurricane Matthew – Haiti, 2016

**Macarthur Charles**[1¤a]*, **Milo Richard**[2], **Mary R. Reichler**[3], **Jean Baptiste Koama**[1], **Willy Morose**[2], **David L. Fitter**[1¤b]

**1** Centers for Disease Control and Prevention, Port-au-Prince, Haiti, **2** Programme National de Lutte contre la Tuberculose (PNLT), Ministère de la Santé Publique et de la Population, Port-au-Prince, Haiti, **3** National Center for HIV/AIDS, Viral Hepatitis, STD and TB Prevention, Centers for Disease Control and Prevention, Atlanta, GA, United States of America

¤a Current Address: Global Tuberculosis Branch, Division of Global HIV and Tuberculosis, Center for Global Health, Centers for Disease Control and Prevention, Atlanta, Georgia, United States of America
¤b Current Address: Immunization Systems Branch, Global Immunization Division, Center for Global Health, Centers for Disease Control and Prevention, Atlanta, Georgia, United States of America
* xzk9@cdc.gov

**Data Availability Statement:** All relevant data are within the manuscript and its Supporting Information files.

## Abstract

### Background

On October 4, 2016, Hurricane Matthew struck southwest Haiti as a category 4 storm. The goal of this study was to evaluate the impact of the hurricane on tuberculosis (TB) services and patient outcomes in the three severely affected departments–Sud, Grand'Anse, and Nippes–of southwest Haiti.

### Methods

We developed a standard questionnaire to assess a convenience sample of health facilities in the affected areas, a patient tracking form, and a line list for tracking all patients with drug-susceptible TB registered in care six months before the hurricane. We analyzed data from the national TB electronic surveillance system to determine outcomes for all patients receiving anti-TB treatment in the affected areas. We used logistic regression analysis to determine factors associated with treatment success.

### Results

Of the 66 health facilities in the three affected departments, we assessed 31, accounting for 536 (45.7%) of 1,174 TB patients registered in care when Hurricane Matthew made landfall in Haiti. Three (9.7%) health facilities sustained moderate to severe damage, whereas 18 (58.1%) were closed for <1 week, and five (16.1%) for $\geq$1 week. Four weeks after the hurricane, 398 (73.1%) of the 536 patients in the assessed facilities were located. Treatment success in the affected departments one year after the hurricane was 81.4%. Receiving care outside the municipality of residence (adjusted odds ratio [aOR]: 0.46, 95% confidence interval [CI]: 0.27–0.80) and HIV positivity (aOR: 0.31, 95% CI: 0.19–0.51) or unknown HIV

**Funding:** The work in this manuscript was supported by the U.S. President's Emergency Plan for AIDS Relief (PEPFAR) through the Centers for Disease Control and Prevention under the terms of GH15-152702/1U2GGH0001643-02.

**Competing interests:** The authors have declared that no competing interests exist.

status (aOR: 0.49, 95% CI: 0.33–0.74) were associated with significantly lower rates of treatment success.

## Conclusions

Despite major challenges, a high percentage of patients receiving anti-TB treatment before the hurricane were located and successfully treated in southwest Haiti. The lessons learned and results presented here may help inform policies and guidelines in similar settings for effective TB control after a natural disaster.

## Introduction

On October 4, 2016, Hurricane Matthew struck southwest Haiti as a category 4 storm, causing widespread devastation and approximately 500 deaths [1]. Three geographic departments were severely affected including Sud, Grand'Anse, and Nippes. The eye of the hurricane went over the town of Les Anglais in Sud and other parts of the Grand'Anse department [2]. An estimated 1.6 million people were affected and over 50,000 were displaced [3]. The three affected departments accounted for ~15% of the total number of notified TB cases in Haiti in 2011–2015 [4]. Early reported problems in the aftermath of the hurricane included the accrued risk for infectious diseases, such as cholera, malaria, and dengue, inaccessibility to safe water and sanitation, and difficulty for patients to access some health facilities, including TB clinics and drug storage facilities [5, 6]. There were also concerns about interruption of TB services, including screening, testing, treatment, and outreach activities [7, 8]. After a natural disaster, cases of TB may go undiagnosed and lead to increased transmission in the community [9, 10]. In addition, patients on treatment may not have access to timely medication and may go on to develop drug resistance resulting in treatment failure.

If hurricanes will become more intense in the future, as is predicted by climate science, understanding and mitigating their impact on TB control will be critical in ensuring progress towards global TB elimination [11–14]. Indeed, some of the countries with the highest TB burden are also vulnerable to natural disasters. Of the 30 high TB burden countries, 27 are considered moderate to very high risk for natural disasters according to the World Risk Report [15, 16]. Yet, there have been relatively few studies on the impact of natural disasters on TB services and outcomes for patients receiving care at the time of the event in high TB burden resource-constrained settings.

The goal of this study was to evaluate the impact of Hurricane Matthew on TB services and patient outcomes in the three severely affected departments of Haiti. The objectives were to (1) assess the impact of the hurricane on TB facilities and services and describe the public health activities implemented to mitigate this impact, (2) assess the impact of the hurricane on treatment outcomes of patients who initiated anti-TB treatment before Hurricane Matthew, and (3) identify patient-related factors associated with treatment success or failure following the hurricane. This study may be of interest to national programs in high TB burden countries vulnerable to natural disasters.

## Materials and methods

### Health facility assessment and study population

We developed a standard questionnaire to assess the impact of the hurricane on TB health facilities and services in the affected areas, a patient tracking form, and a line list to help health

providers account for all patients who were registered for TB care six months before the hurricane (S1 and S2 Appendices). We selected a convenience sample consisting TB facilities in each of the three affected departments based on accessibility by roads, security concerns, and availability of providers. We assessed the level of damage to TB clinic infrastructure, the condition of diagnostic equipment, case registers and treatment cards, the status of the drug supply, and the availability of staff. At each clinic, we prepared line lists with the providers for all TB patients on treatment prior to the event. Patients were designated as located or not located based on interview of TB providers and review of TB registers and/or treatment cards, where available.

To assess the impact of the hurricane on treatment outcomes of patients who initiated anti-TB treatment before Hurricane Matthew, we conducted a retrospective study using data from the national TB electronic surveillance system. All patients receiving anti-TB treatment in all health facilities in the three affected departments–Sud, Grand'Anse, and Nippes–during the six months, from April 5 to September 30, 2016 (quarters 2 and 3), preceding Hurricane Matthew were included (Fig 1). In accordance with Haiti's Programme National de Lutte contre la Tuberculose (PNLT) guidelines, anti-TB treatment consisted of two months of Isoniazid, Rifampin, Pyrazinamide, and Ethambutol during the intensive phase of treatment followed by four months of Isoniazid and Rifampin during the maintenance phase. Patients received medication for 2–4 weeks in the intensive phase of treatment and for 4–8 weeks during the maintenance phase. As a contingency measure to prevent treatment interruptions, health care providers routinely ensured that patients had an additional 1–2 weeks of medications in hand beyond their scheduled follow-up visits.

In our study, patients who started treatment in quarters 2 and 3 represented the exposure group since treatment started in those quarters could have been disrupted because it would

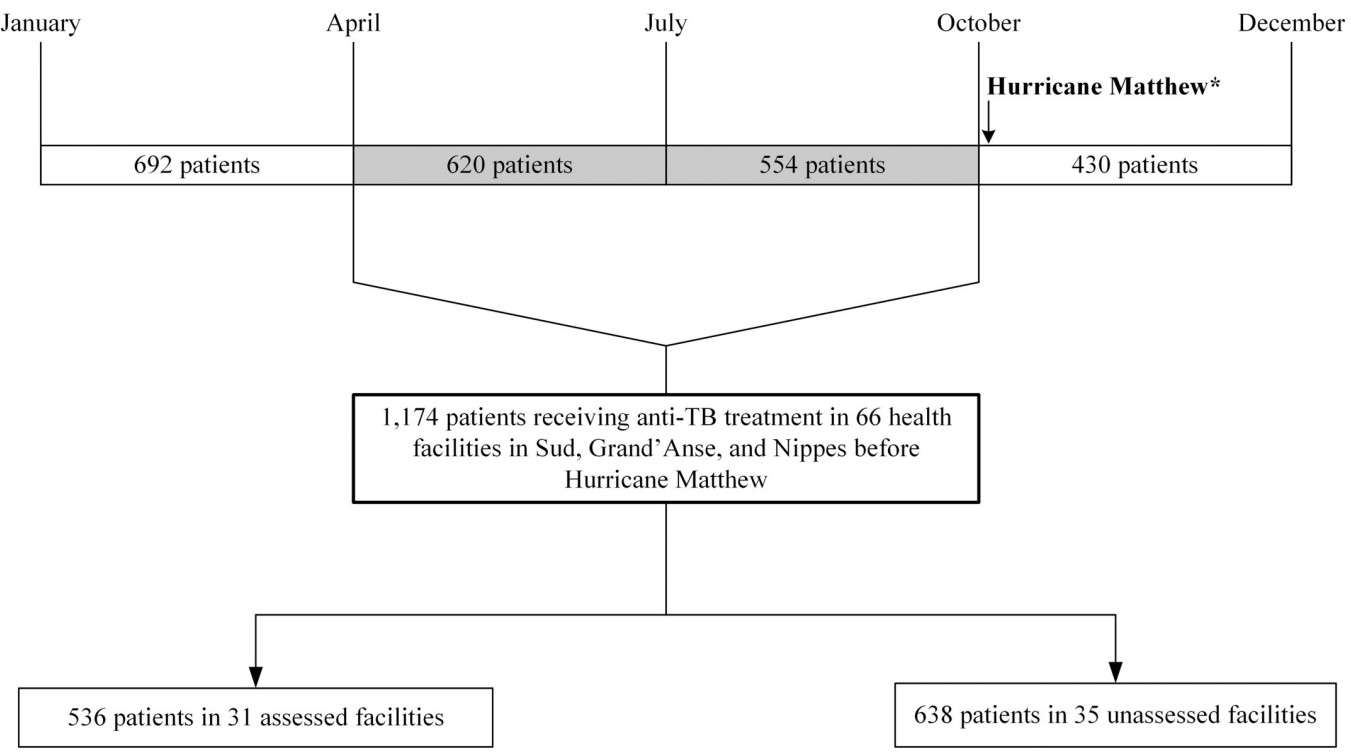

**Fig 1. Timeline of TB patient enrollment in care and study population, Haiti, 2016.** The shaded area covers the period from April 5 to September 30, 2016, during which the study population initiated anti-TB treatment. *Hurricane Matthew made landfall in Haiti on October 4, 2016.

not have been completed until after the hurricane occurred. Patients who initiated anti-TB treatment in quarter 1 (before the hurricane) and quarter 4 (after the hurricane) would not be disrupted by the hurricane (unless the treatment was prolonged) and therefore represented comparison groups for treatment outcomes. We compared treatment outcomes for the three affected departments and for each quarter in order to determine whether Hurricane Matthew had an impact on treatment success and loss to follow-up.

## Data collection

We used Microsoft Excel (Redmond, WA, USA) to compile data from the assessment and patient tracking forms. In addition, PNLT provided anonymized data of all TB cases reported in 2016 from the electronic TB surveillance system [4, 17]. The data included geographic sub-divisions (communes or municipalities) of Haiti's ten departments and specific health facilities where patients received anti-TB treatment, patient demographics, TB treatment history, sputum microscopy, type of TB disease, human immunodeficiency virus infection (HIV) status, and TB treatment outcome.

## Data analysis

We used STATA, version 15 (Stata Corp, College Station, TX, USA) for analysis. Standard World Health Organization definitions of TB case notification and treatment outcomes were used in this study [18]. The primary outcome was treatment success, which is defined as the proportion of TB cases successfully treated (cured or completed treatment) among all TB cases notified to PNLT from April 5 to September 30, 2016 [18]. Adverse treatment outcomes included lost to follow up, died during treatment, not evaluated, and treatment failed.

We used Pearson's chi-squared test to determine statistical differences between characteristics of patients in the assessed health facilities and those in the unassessed facilities. We used survey statistical procedures to account for possible site-level clustering before performing the logistic regression analysis. TB facilities with similar numbers of patients were considered a cluster. We conducted bivariable and multivariable logistic regression analysis to determine associations between epidemiologically relevant factors (age, gender, commune of residence, smear status, category of patient, and HIV status) and treatment success [19–22]. A p-value <0.05 was considered statistically significant.

## Ethics statement

The Haiti National Bioethics Committee determined this project to be a public health program evaluation and did not require review by the Institutional Review Board. This project was reviewed in accordance with the Centers for Disease Control and Prevention (CDC) human research protection procedures and was determined to be non-research, epidemic disease control activity. The requirement for informed consent was waived and the patient records used in our retrospective study were anonymized data from the national TB electronic surveillance system.

## Results

### Assessment of TB facilities in Sud, Grand'Anse, and Nippes after Hurricane Matthew

We assessed 31 (47.0%) of the 66 health facilities in the affected departments within an average of 21 (range 15–25) days after the hurricane. These 31 facilities accounted for 536 (45.7%) of the 1,174 patients in TB care in the three departments from April 5 to September 30, 2016, just

before Hurricane Matthew made landfall (Fig 1): 20 facilities in Sud with 224 (19.1%) patients, 7 in Grand'Anse with 259 (22.1%) patients, and 4 in Nippes with 53 (4.5%) patients. The 35 unassessed facilities accounted for 638 (54.3%) of the 1,174 patients: 15 in Sud with 345 (29.4%) patients, 10 in Grand'Anse with 133 (11.3%) patients, and 10 in Nippes with 160 (13.6%) patients.

Among the 31 assessed health facilities, 28 (90.3%) suffered minimal to no damage. This included the 20 facilities in Sud, the four in Nippes four in Grand'Anse. Three (9.7%) health facilities in Grand'Anse sustained moderate to severe damage, including loss of roof, wall collapse, and flooding resulting in loss of equipment and materials. Eight (25.8%) facilities had no service interruption, whereas 18 (58.1%) were closed for less than one week, and five (16.1%) were closed for one week or longer. TB case registers were intact at 26 (83.9%) health facilities but were damaged or lost at 5 (16.1%) others, whereas microscopes were intact at 27 (87.1%) facilities and damaged at 4 (12.9%) facilities. The drug supply was adequate at all health facilities except for one clinic in Sud, which experienced a shortage, but this was corrected within three days. No patient receiving care at this facility had treatment interruption as a result of this shortage.

Four weeks after the hurricane, 398 of the 536 (73.1%) patients in the 31 assessed facilities were located: 277 (69.6%) patients returned on their own, 90 (22.6%) were reached through phone calls, and 31 (7.8%) after contacting relatives or friends. Of these located patients, 339 (85.2%) had no treatment interruption, 52 (13.1%) had interruption of less than one week, and 7 (1.8%) had interruption of 1–2 weeks. Of the 153 patients who were not yet located during our assessment, 56 (36.6%) had been lost to follow up before the hurricane, 32 (20.9%) had missed their scheduled appointments, and 65 (42.5%) were not yet due for their regularly scheduled appointments since the hurricane. All patients who returned for a follow up visit had access to medications. A total of 113 new patients were diagnosed with TB at the 31 facilities in the four weeks after the hurricane, whereas the year before (2015), 193 new cases were diagnosed in the same period. Some of the major challenges facing TB patients and TB care providers after the hurricane included lack of shelter, food insecurity, scarce potable water, and persistent torrential rain resulting in flooding, mudslides, and impassable roads.

### Description of the study population

A total of 1,174 patients were registered for anti-TB treatment from April 5 to September 30, 2016 in the three departments when Hurricane Matthew made landfall in Haiti. There were 536 (45.7%) patients in the facilities we assessed and 638 (54.3%) patients in the facilities we did not assess (Fig 1). Of the 1,174 patients, 46.4% were female. The median age was 31 years (interquartile range [IQR]: 23–44). Over half (53.9%) of the patients were in the 15–34 age group, whereas children 14 years old and younger comprised 4.4% of the study population (Table 1). Overall, 91.8% of the patients lived in the same commune as the facility where they received TB care and 97.9% had a family member or friend as an *accompagnateur* (companion) to ensure they are compliant with treatment in accordance with PNLT's directly observed therapy guidelines. Furthermore, most of the patients (93.7%) were classified as new, 81.2%% had smear-positive pulmonary TB, 9.5% had smear negative pulmonary TB, and 9.3% had extra-pulmonary TB. In terms of HIV status, 901 (76.8%) had a negative result, 123 (10.5%) were HIV-positive, and 150 (12.8%) had an unknown status; antiretroviral therapy was documented for 78 (63.4%) of the HIV-positive patients.

### Treatment outcomes

The treatment success rate for the 1,174 patients receiving TB care before Hurricane Matthew in the three affected departments was 81.4% and the loss to follow up rate was 10.2% (Table 2).

**Table 1. Characteristics of the 1,174 patients receiving TB treatment in the affected departments before Hurricane Matthew.**

| Characteristics | Assessed (N = 536) | Not Assessed (N = 638) | All TB Facilities (N = 1,174) |
|---|---|---|---|
| | n (%) | n (%) | n (%) |
| **Department** | | | |
| Sud | 224 (41.8) | 345 (54.1) | 569 (48.5)*** |
| Grand'Anse | 259 (48.3) | 133 (20.8) | 392 (33.4) |
| Nippes | 53 (9.9) | 160 (25.1) | 213 (18.1) |
| **Gender** | | | |
| Female | 253 (47.2) | 292 (45.8) | 545 (46.4) |
| Male | 283 (52.8) | 346 (54.2) | 629 (53.6) |
| **Median age (IQR)** | 30 (23–44.5) | 31 (23–43) | 31 (23–44) |
| **Age group** | | | |
| 0–14 years old | 24 (4.5) | 28 (4.4) | 52 (4.4) |
| 15–34 years old | 288 (53.7) | 346 (54.2) | 634 (54.0) |
| 35–54 years old | 146 (27.2) | 177 (27.7) | 323 (27.5) |
| 55 years and older | 78 (14.6) | 87 (13.6) | 165 (14.1) |
| **Commune of residence** | | | |
| Same as facility | 470 (87.7) | 608 (95.3) | 1,078 (91.8)*** |
| Different | 66 (12.3) | 30 (4.7) | 96 (8.2) |
| *Accompagnateur*† | | | |
| Family/Friend | 524 (97.8) | 625 (98.0) | 1,149 (97.9) |
| Health agent/other | 12 (2.2) | 13 (2.0) | 25 (2.1) |
| **Patient category** | | | |
| New | 491 (91.6) | 609 (95.5) | 1,100 (93.7)** |
| Previously treated§ | 45 (8.4) | 29 (4.5) | 74 (6.3) |
| **Site of TB disease and smear status** | | | |
| Pulmonary smear-positive | 468 (87.3) | 485 (76.0) | 953 (81.2)*** |
| Pulmonary smear-negative | 46 (8.6) | 64 (10.0) | 110 (9.4) |
| Extrapulmonary | 22 (4.1) | 89 (13.9) | 111 (9.5) |
| **HIV status** | | | |
| Negative | 393 (73.3) | 508 (79.6) | 901 (76.7)* |
| Positive | 61 (11.4) | 62 (9.7) | 123 (10.5) |
| Unknown | 82 (15.3) | 68 (10.7) | 150 (12.8) |

†*Accompagnateur*: A companion, usually a family member or friend, who ensures that the patient remains compliant with treatment.

Previously treated includes the following patient categories: relapse, failure, and resumption of treatment.

*: p-value <0.05;

**: p-value <0.01;

***: p-value <0.001

Nippes had the highest treatment success rate at 87.3% and lowest loss to follow up rate (7.0%). Sud and Grand'Anse had treatment success rates of 81.3% and 78.4%, respectively, with loss to follow up rates of 9.3% and 13.3%, respectively.

Treatment success for the 536 patients receiving TB treatment before Hurricane Matthew in the assessed facilities was lower than for the 638 patients who were treated in the unassessed facilities (76.9% vs. 85.3%; p-value <0.001). In assessed facilities, treatment success fell from 82.4% in the first quarter to 79.8% in the second quarter and to 73.1% in the third quarter, whereas it remained stable in unassessed facilities (84.0–86.8%) (Fig 2). In addition, among assessed facilities, those with moderate to severe structural damage were significantly more

**Table 2. Outcomes for patients receiving anti-TB treatment in the affected departments before Hurricane Matthew.**

| Treatment category | Sud (N = 569) | Grand'Anse (N = 392) | Nippes (N = 213) | Total (N = 1,174) |
|---|---|---|---|---|
| Cured | 328 (57.6) | 237 (60.5) | 151 (70.9) | 716 (61.0)* |
| Completed treatment | 135 (23.7) | 70 (17.9) | 35 (16.4) | 240 (20.4) |
| Lost to follow up | 53 (9.3) | 52 (13.3) | 15 (7.0) | 120 (10.2) |
| Died | 37 (6.5) | 24 (6.1) | 7 (3.3) | 68 (5.8) |
| Treatment failed | 5 (0.9) | 6 (1.5) | 2 (0.9) | 13 (1.1) |
| Not evaluated | 11 (1.9) | 3 (0.7) | 3 (1.4) | 17 (1.4) |
| Successfully treated† | 463 (81.3) | 307 (78.4) | 186 (87.3) | 956 (81.4) |

†Successfully treated = the sum of treatment completed and cured.

*p-value <0.05

likely to experience lower treatment success rates after the hurricane (O.R.: 1.92; 95% C.I.: 1.14–3.24; p-value = 0.015).

Patients who initiated anti-TB treatment in Apr-Jun (quarter 2) and in Jul-Sep (quarter 3) represent the study population. Those who initiated treatment in Jan-Mar (quarter 1) and in Oct-Dec (quarter 4) represent the comparison groups for treatment success.

**Factors associated with treatment success.** Bivariable analysis of patient demographic and clinical characteristics (Table 3) showed that receiving care outside of the commune of residence, having a history of previous TB treatment, and being either HIV-positive or of unknown HIV status were associated with lower treatment success rates. In the multivariable analysis, receiving care outside of the commune of residence and HIV positivity or unknown HIV status were associated with significantly lower rates of treatment success.

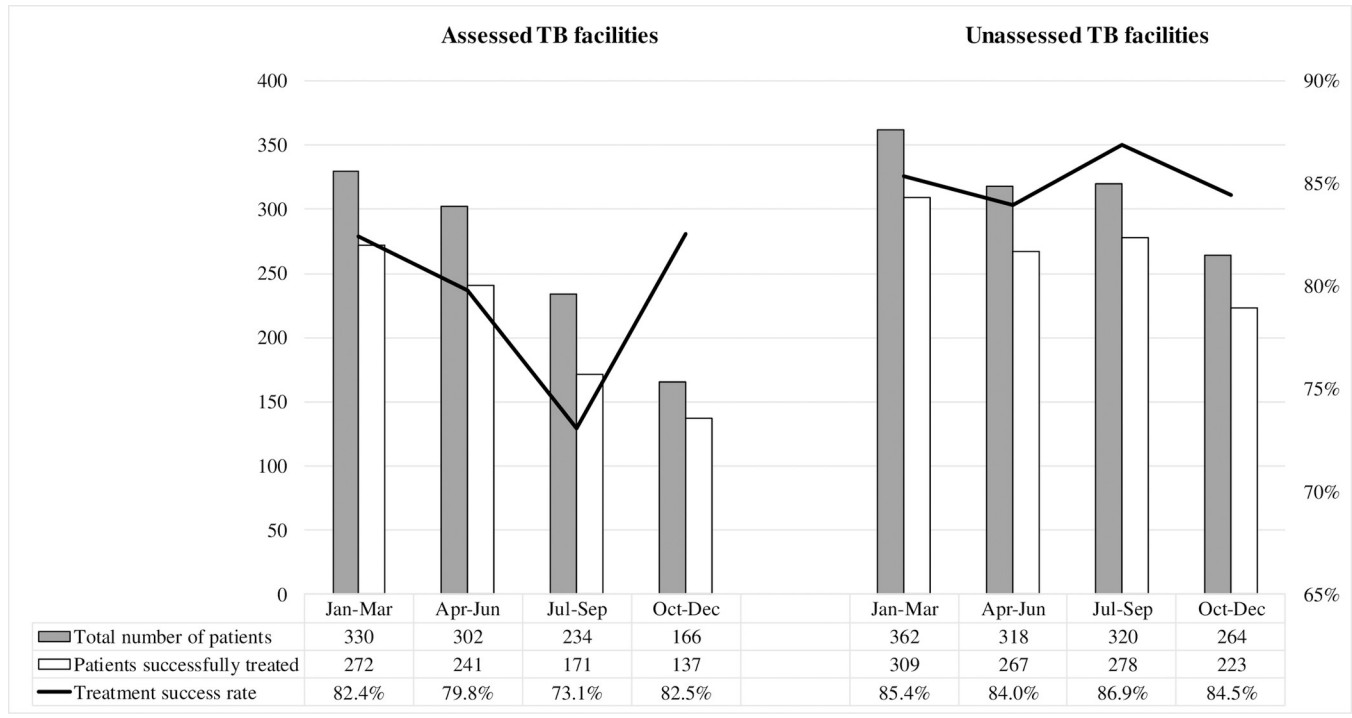

**Fig 2. Treatment success by quarter according to whether facilities were assessed or not, Haiti, 2016.**

**Table 3. Factors associated with treatment success in patients receiving care in health facilities affected by Hurricane Matthew.**

| Background Characteristic | Bivariable analysis | Multivariable analysis |
|---|---|---|
| | OR† (95% CI#) | aOR‡ (95% CI) |
| **Age (years)** | | |
| 0–14 | 0.73 (0.34–1.56) | 0.86 (0.34–2.23) |
| 15–34 | Reference | Reference |
| 35–54 | 0.75 (0.51–1.10) | 0.96 (0.63–1.45) |
| 55+ | 0.70 (0.39–1.27) | 0.76 (0.42–1.36) |
| **Gender** | | |
| Female | Reference | Reference |
| Male | 0.93 (0.69–1.26) | 0.87 (0.64–1.19) |
| **Commune of residence** | | |
| Same as facility | Reference | Reference |
| Different from facility | 0.46 (0.22–0.99)* | 0.46 (0.27–0.80)** |
| **Smear status** | | |
| Positive | Reference | Reference |
| Negative | 0.70 (0.30–1.65) | 0.80 (0.36–1.77) |
| Not done | 0.80 (0.37–1.73) | 0.82 (0.39–1.72) |
| **Patient type** | | |
| New patient | Reference | Reference |
| Previously treated§ | 0.45 (0.25–0.78)** | 0.55 (0.29–1.04) |
| **HIV status** | | |
| Negative | Reference | Reference |
| Positive | 0.30 (0.19–0.47)*** | 0.31 (0.19–0.51)*** |
| Unknown | 0.48 (0.31–0.73)** | 0.49 (0.33–0.74)** |

†OR = odds ratio; #CI = confidence interval;

‡aOR = adjusted odds ratio

§Previously treated includes the following patient categories: relapse, failure, and resumption of treatment.

*: p-value <0.05;

**: p-value <0.01;

***: p-value <0.001

## Discussion

Our study shows an overall treatment success rate of 81% for patients receiving anti-TB treatment before Hurricane Matthew in the three departments most severely affected. This is comparable to the treatment success rates for these three departments in 2011–2015 and to the national treatment success rate of 82% in 2016 [4, 23]. In addition, 73% of the patients receiving care at the assessed facilities were located within four weeks of Hurricane Matthew and 98% of the located patients experienced little to no treatment interruption. This is remarkable given that Matthew, as the first category 4 hurricane to strike Haiti in 52 years, dealt a major setback to the southwest region. The dedication of TB health providers in the affected departments and the heroic steps they took to ensure that patients were accounted for after the hurricane were exemplary. Health providers ensured patients had surplus medications before the hurricane, especially those who lived far from the health facilities. After the hurricane, they phoned patients and their contacts (relatives or friends) repeatedly; when that failed, they visited patients or invited them to their homes or met them at specific landmarks. Some providers also relied on patients whom they had successfully reached to contact the ones who had not

yet returned. We also observed across many of the facilities we visited, especially the more rural ones, that health providers knew their patients well and could even recall some patients without consulting the registers. The high rate of patients returning to health facilities on their own suggests that patients had received appropriate adherence counseling and that, despite the dire circumstances, they still returned for care [24]. This was also observed in a post-typhoon Haiyan evaluation of the TB program in the Philippines [25].

We found that patients who initiated treatment in the second and third quarters of 2016 before Hurricane Matthew had higher loss to follow up rates than those who initiated treatment in the first and last quarters. This reflects the immediate migration of patients out of the affected areas in the aftermath of the disaster. However, the overall treatment success rate for the affected areas was not adversely affected by the transient increase in loss to follow up.

The lower treatment success in the assessed facilities could be due to a greater percentage of patients living outside the communes of these facilities. Indeed, we know that larger referral facilities tend to have lower treatment success rates as a result of higher loss to follow up rates [26]. This suggests these patients would have had to travel farther for care or would have had more difficulty coming up with the funds necessary for taking public transport to their visits and would have had, as a result, a lower risk of achieving treatment success. This is corroborated by our finding that receiving treatment in facilities located outside the communes of residence was associated with significantly lower rates of treatment success. Indeed, transportation cost to health facilities can be a major impediment to seeking care [27–31]. Therefore, PNLT should continue to strengthen efforts to appropriately refer patients who reside outside of their communes to health facilities closer to their domiciles.

Among assessed facilities, those that sustained moderate to severe damage also had lower treatment success rates after the hurricane compared to those that had no or only minor damage. This is not surprising as patients may have been less inclined to return to facilities that they knew had experienced major damage.

In addition, assessed facilities had a greater proportion of HIV-positive patients or those of unknown HIV status, which could also help explain the lower anti-TB treatment success observed in these facilities. Our study found that HIV-positive patients or those of unknown HIV status were significantly less likely to achieve treatment success. This result suggests that HIV testing and/or initiation of antiretroviral therapy was likely delayed for patients with TB. At the time of this study, antiretroviral therapy was not available at one of the largest TB facilities in the Grand'Anse department. HIV-positive TB patients at this facility had to be referred to a nearby hospital for antiretroviral therapy. Globally, in 2019, only 69% of notified TB patients had a documented HIV test result and anti-TB treatment success was 76% for HIV-positive TB patients compared to 85% for those who are HIV negative [15]. Lower treatment success for HIV-positive TB patients may be due to delayed TB diagnosis, delayed HIV testing and/or initiation of ART, and challenges in treating both conditions simultaneously (higher pill burden, side effects, and drug interactions) [32–34]. Greater efforts should be directed in the Southwest region of Haiti toward ensuring that TB patients receive HIV counseling and testing promptly and, if HIV-positive, referred for antiretroviral therapy as soon as possible. Likewise, HIV-positive patients should be systematically screened for TB symptoms and, if screening is positive, referred for testing by the Xpert MTB/RIF assay [35, 36]. Patient-centered interventions focused on strengthening TB/HIV collaboration, providing adherence support, and tracing of patients lost to follow up are needed to increase treatment success among patients with both conditions [37, 38].

Even though some health facilities were intact, fewer new patients were coming for TB screening in the aftermath of the hurricane. This is understandable because new patients may not have realized some facilities were spared and may have had more pressing needs after the

disaster than visiting a health facility for illness. Therefore, efforts to promote the full function of TB diagnostic and treatment services at facilities should be considered so that all persons meeting the case definition for suspect TB present to the TB facilities for evaluation. In addition, active case finding activities such as screening populations in temporary shelters post-event or persons residing in slum areas may also be considered to increase TB case detection [35, 39].

Our study has some limitations. We chose to assess facilities which we thought could be more severely affected, several of which were referral facilities that were easily accessible along major roads. We could not assess some facilities due to recurrent flooding and/or security concerns. Therefore, this approach could have resulted in selection bias. In addition, given the retrospective design, we had to limit the analysis to only those variables available in the TB surveillance database. We could not explore in our analysis other factors–weight, smoking history, social factors (loss of income, loss of home or loss of family members), and other comorbidities such as diabetes mellitus–which could have impacted treatment success.

In conclusion, our study shows that, despite major challenges, a high percentage of patients receiving anti-TB treatment before the hurricane were located and successfully treated in southwest Haiti. Our findings may be relevant for other resource-limited settings with a high TB burden and vulnerable to natural disasters. Factors that contributed to mitigating the impact of the hurricane on anti-TB treatment success included multiple layers of redundancy in the collection of TB surveillance data, dedicated health providers, strong patient education/support, surplus medications beyond scheduled visits, and a regularly updated central electronic TB database. The lessons learned and results presented here may help inform policies and guidelines to establish mitigation practices in similar settings for effective TB control after a natural disaster.

## Supporting information

**S1 Appendix. TB health facility assessment form.**
(DOCX)

**S2 Appendix. TB patient tracking form and line list.**
(DOCX)

**S1 Data.**
(XLSX)

## Acknowledgments

We express our deepest gratitude to the TB health care providers in the affected areas–Sud, Grand'Anse, and Nippes–who took extraordinary steps to ensure that patients were accounted for and had medications available to continue with treatment. We are also grateful to the Logistics Team at the Centers for Disease Control and Prevention in Haiti, especially the drivers, for their unfaltering support during this difficult period.

## Author Contributions

**Conceptualization:** Macarthur Charles, Milo Richard, Mary R. Reichler, Jean Baptiste Koama, Willy Morose, David L. Fitter.

**Data curation:** Macarthur Charles.

**Formal analysis:** Macarthur Charles.

**Investigation:** Mary R. Reichler.

**Methodology:** Milo Richard, Jean Baptiste Koama.

**Project administration:** Macarthur Charles.

**Supervision:** Willy Morose, David L. Fitter.

**Writing – original draft:** Macarthur Charles.

**Writing – review & editing:** Macarthur Charles, Milo Richard, Mary R. Reichler, Jean Baptiste Koama, Willy Morose, David L. Fitter.

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
