## [Decision Letter · Decision Letter 0]

6 Jul 2020

PONE-D-20-06687

Treatment success for patients with tuberculosis receiving care in areas severely affected by Hurricane Matthew – Haiti, 2016

PLOS ONE

Dear Dr. Charles,

Thank you for submitting your manuscript to PLOS ONE. After careful consideration, we feel that it has merit but does not fully meet PLOS ONE’s publication criteria as it currently stands. Therefore, we invite you to submit a revised version of the manuscript that addresses the points raised during the review process.

We look forward to receiving your revised manuscript.

Kind regards,

Anna Maria Mandalakas, MD

Academic Editor

PLOS ONE

Journal Requirements:

2. In ethics statement in the manuscript and in the online submission form, please provide additional information about the patient records/samples used in your retrospective study. Specifically, please ensure that you have discussed whether all data/samples were fully anonymized before you accessed them and/or whether the IRB or ethics committee waived the requirement for informed consent. If patients provided informed written consent to have data/samples from their medical records used in research, please include this information.

3. Please include additional information regarding the survey or questionnaire used in the study and ensure that you have provided sufficient details that others could replicate the analyses. If you developed and/or translated a questionnaire as part of this study and it is not under a copyright license more restrictive than Creative Commons Attribution (CC-BY), please include a copy, in both the original language and English, as Supporting Information.

Additional Editor Comments (if provided):

Thank you for submitting your manuscript to Plos One. The authors should be commended on their work which could potentially add to the paucity of data regarding the impact of natural disasters on TB control. Nevertheless, based on editorial and external review, the manuscript is requires major revision before it can be considered further for publication.

In addition to responding to the useful suggestions provided by the external reviewers, the manuscript may be strengthen by consideration of the following recent publication:

Prevalence of Tuberculosis in Children After Natural Disasters, Bohol, Philippines.

Murray KO, Castillo-Carandang NT, Mandalakas AM, Cruz AT, Leining LM, Gatchalian SR; PEER Health Bohol Pediatric Study Team.

Emerg Infect Dis. 2019 Oct;25(10):1884-1892. doi: 10.3201/eid2510.190619.

PMID: 31538561

Reviewers' comments:

Reviewer's Responses to Questions

**Comments to the Author**

1. Is the manuscript technically sound, and do the data support the conclusions?

Reviewer #1: Yes

Reviewer #2: Yes

2. Has the statistical analysis been performed appropriately and rigorously? 

Reviewer #1: Yes

Reviewer #2: Yes

3. Have the authors made all data underlying the findings in their manuscript fully available?

Reviewer #1: Yes

Reviewer #2: Yes

4. Is the manuscript presented in an intelligible fashion and written in standard English?

Reviewer #1: Yes

Reviewer #2: Yes

5. Review Comments to the Author

Reviewer #1: Dear authors,

Thank you for the opportunity to review your work. This study of treatment success for patients with drug-sensitive TB at the time of Hurricane Matthew has value to help communities develop disaster resilience for TB treatment as well as learn how to best respond after a disaster. There is a wealth of data available and most of it is well presented, but there are changes that could be made to make the results of the study more useful for readers:

1. With the relative success in treating TB despite the damage from Hurricane Matthew, it is important to describe the aspects of facilities that maintained equivalent treatment success both before and after the hurricane. Therefore, since details about unassessed facilities are unknown, the authors should delve into the different characteristics of the assessed facilities and do analyses to assess how variable aspects of the facilities and providers contributed to individual facility treatment success rates. Assessed facilities should be divided into those with equivalent treatment rates before and after the hurricane, and those with lower rates after the hurricane. Then, analyses should be done to determine if damage to the facility or other assessed factors correlated with the treatment success rate.

2. 69: Please explain why patients were included whose treatment started April 1-4, since those patients should have had their treatment completed prior to the hurricane. Please also address this in the discussion and limitations, since it affects the results.

3. 79-81: S1 Appendix does not show the form used to assess the facility; it is the patient tracking form. Would the authors please include the form for facility assessment? Please also keep the patient tracking form as an appendix, as well as the line item form.

4. 157: Please clarify for how much time there was a drug shortage. It would also help the reader to know how many days’ worth of medications each patient was given at every clinic visit and how often they were expected to return for follow-up.

5. 173: Was the TB success rate determined only from PNLT data or from individual facility records, as well? When was the data accessed to determine treatment success? This would be prospective for most cases, since the target patients were identified within a few weeks after the hurricane and then followed to see if they were treated successfully. Was there a possibility that data entry into PNLT was delayed and results would have been different if the database had been accessed later?

6. 188-189: The difference could also have been due to treatment failure or not completing treatment, correct? Please clarify.

7. 200 and Table 3: Please describe what was included in the multivariable analysis.

8. 221-222: Please explain the specific actions taken by TB providers. This will help readers consider how this could be replicated in similar settings.

9. 234-235: If you choose to keep the comparison between assessed and unassessed facilities (the deeper dive into assessed facilities seems most important but this could be valuable as well), there is a need to further elucidate their differences. In other words, were assessed facilities larger than unassessed? Did they have more people from outside the commune of residence? What other aspects differed between the groups?

10. 258-260: The intent of this sentence is not clear. Would you kindly rewrite it to convey the information in a different way?

11. In your conclusion, please provide specific recommendations (based on what was learned from this study) on how facilities and providers of TB care can be better prepared in case of future hurricanes or other disasters, acknowledging the limits of generalizability.

12. Beyond the outcome of treatment success, is there data available on how many of the patients experienced interruptions in treatment? If so, for how long were the interruptions? Were any at least 2 weeks in duration? Is so, was it the initiation or continuation phase? Did any develop drug resistance? These outcomes are important to report, if possible.

13. In general, please scrutinize the grammar in detail as there are a few typos. In addition, please include the bracketed references before the period at the end of the sentence being referenced.

14. What are the possible reasons why patients with HIV or unknown HIV status had lower treatment success? Was this different before and after the hurricane or the same throughout?

Thank you for undertaking this research and putting together a great paper. With minor revisions, this would be a helpful addition to the literature.

Reviewer #2: Comments to the author:

General:

This is an interesting and well-done study in a low resource setting describing an impressive and successful public health activity and highlighting the important relationship between natural disasters and TB control. If natural disasters become more prevalent, as is predicted by climate change science, understanding and mitigating the impacts of these disasters on TB control will be critical in ensuring progress towards global TB elimination.

Major comments:

(1) better clarify the goals and objectives of the project and study and reformat the sections based on these goals and objectives,

(2) build up the discussion section further, and emphasize the significance and applicability of your findings to other similar settings outside of Haiti.

Specific comments:

Introduction

• Lines 56-60 seem out of place and doesn’t really fit in with the flow of the introduction. Specifically, it seems that in this paragraph, the authors are outlining the argument of why TB outcomes may worsen following a natural disaster. Citing WHO guidelines on restarting TB treatment after interruption doesn’t seem to fit with this argument.

• Lines 61-63 The goal doesn’t seem to me as a goal but rather the activities you conducted in this important public heath surveillance project. It seems that the goal of this project was to evaluate the impact that hurricane Matthew had on TB services and outcomes in the three departments of Haiti. After you identify this goal, then you can delineate the specific objectives of this manuscript. It seems that the objectives were to (1) assess the impact of the hurricane on TB facilities and services and describe the public health activities implemented to mitigate this impact, (2) assess the impact of the hurricane on treatment outcomes of those who were linked to TB therapy prior to the disaster, and (3) identify individual-related factors associated with treatment success or failure following the hurricane.

• I would add a sentence emphasizing the importance of this study to the general reader, i.e. why are the findings important. I would focus on how many low resource settings are struck by both high TB prevalence and preponderance of devastating natural disasters and understanding how these phenomena are related can inform TB control activities.

Methods

• Lines 73-75 I would be more specific on how this comparison is a representation of the impact of the hurricane on treatment outcomes. You could say something like “treatment started in quarter 1 would probably not be disrupted by the hurricane (unless the treatment was prolonged) and therefore represents a comparison group. While treatment started in quarters 2 and 3 represents the exposure group since treatment started in those quarters would have been disrupted since it would not have been completed until after the hurricane occurred.” I think this will make it clearer how you plan to compare treatment success before and after the hurricane.

• Lines 99-100 Please include citation for WHO definitions

• Lines 105-107 Did you use both chi square tests and odd ratios to determine association between risk factors and success. I would reword this sentence by saying you conducted bivariate analysis using odds ratios (and or chi square) with p-value<0.05.

• Lines 108-109 I am confused about the order of things in your analysis. It seems that from your results that you chose the appropriate epidemiologically relevant factors and conducted both bivariable and multivariable analysis on the same factors. If this is the case I would specifically say that.

Results

• Table 1 Should include chi-square testing with p-values to determine if there were statistical differences between those subjects assessed and those that weren’t assessed

• It would be interesting to include estimates of treatment success

• Table 2 Should include chi-square testing with p-values to determine if there were statistical differences between treatment outcomes among the different sites. Also I would reword the title emphasizing that these are outcomes prior to the hurricane (i.e. Treatment initiated in quarters 1-3 before hurricane Matthew).

• Lines 186-189 This assertion fits better in the discussion, specifically in the paragraph lines 229-233

• Table 3 I would eliminate the columns with p-values and denote the statistical significance of the ORs with asterisks. For example * for p-value<0.05 and ** for p-value<0.01.

Discussion

• Lines 225-228 Please include citation, whether it is a publication or based on anecdotal discussions with stakeholders

• Lines 229-233 I know you mention in the introduction how the hurricane primarily stuck the areas that were assessed. But I would further describe how the geography of the hurricane, and perhaps other factors explain your findings, specifically that those in the assessed areas had a higher chance of treatment failure than those in areas that weren’t assessed. Perhaps you could use geospatial analysis to illustrate where the hurricane hit hardest and the percentage of treatment success in those areas.

• Lines 241-243, I would try to identify references or data from other settings that further explore the financial and travel-related restrictions placed on patients who live far from their healthcare facilities. This will add strength to your argument and perhaps improve the generalizability of your findings to other post-disaster settings.

• Can you also comment on the role of HIV status on TB treatment success. You identify it as a statistically significant factor in the results but don’t mention it in the discussion.

• Can you discuss further generalizability of your findings to other post disaster settings and possible limitation of applying your findings to other settings. An important take away readers will likely want to make from this study is how do these findings compare with those of other low resource, high burden Tb settings struck by natural disasters and how your findings could be used to inform mitigation efforts after future disasters.

6. PLOS authors have the option to publish the peer review history of their article (what does this mean?). If published, this will include your full peer review and any attached files.

Reviewer #1: No

Reviewer #2: No

---

## [Author Response · Author response to Decision Letter 0]

15 Jan 2021

Comments from Academic Editor:

Response: Thank you for the links. We have reviewed the contents closely to ensure our manuscript meets PLOS ONE’s style requirements, including those for file naming.

2. In ethics statement in the manuscript and in the online submission form, please provide additional information about the patient records/samples used in your retrospective study. Specifically, please ensure that you have discussed whether all data/samples were fully anonymized before you accessed them and/or whether the IRB or ethics committee waived the requirement for informed consent. If patients provided informed written consent to have data/samples from their medical records used in research, please include this information.

Response: We thank you for pointing this out. We now state in the Ethics statement section that the patient records used in our retrospective study were anonymized data from the national TB electronic surveillance system. We also address in this section that the Haiti National Bioethics Committee and the Centers for Disease Control and Prevention determined this project to be a non-research, epidemic disease control evaluation. The requirement for informed consent was waived.

 3. Please include additional information regarding the survey or questionnaire used in the study and ensure that you have provided sufficient details that others could replicate the analyses. If you developed and/or translated a questionnaire as part of this study and it is not under a copyright license more restrictive than Creative Commons Attribution (CC-BY), please include a copy, in both the original language and English, as Supporting Information.

Response: All questionnaires are now included in the Supporting Information as S1 Appendix. TB health facility assessment form and S2 Appendix. Patient tracking form and line list.

Response: We have now included the captions for the Supporting Information files at the end of the manuscript. We have updated in-text citations accordingly.

Comments from Reviewer 1: 

1. With the relative success in treating TB despite the damage from Hurricane Matthew, it is important to describe the aspects of facilities that maintained equivalent treatment success both before and after the hurricane. Therefore, since details about unassessed facilities are unknown, the authors should delve into the different characteristics of the assessed facilities and do analyses to assess how variable aspects of the facilities and providers contributed to individual facility treatment success rates. Assessed facilities should be divided into those with equivalent treatment rates before and after the hurricane, and those with lower rates after the hurricane. Then, analyses should be done to determine if damage to the facility or other assessed factors correlated with the treatment success rate.

Response: We thank Reviewer 1 for the suggestion. We agree that it is important to describe aspects of assessed facilities that maintained equivalent treatment success rates before and after the hurricane compared to those with lower rates after the hurricane. We examined factors associated with a lower treatment success rate after the hurricane in the assessed facilities. We found that assessed facilities with moderate to severe damage were significantly more likely to experience a lower treatment success rate after the hurricane (Odds ratio: 1.92; 95% confidence interval: 1.14-3.24; p = 0.015). This finding has been added to the Results section (lines 217-219, page 11).

2. Line 69: Please explain why patients were included whose treatment started April 1-4, since those patients should have had their treatment completed prior to the hurricane. Please also address this in the discussion and limitations, since it affects the results.

Response: We thank Reviewer 1 for pointing this out. We have excluded 27 patients who started treatment April 1-4. The total number of patients in our study is now 1,174. We have revised the manuscript accordingly.

3. 79-81: S1 Appendix does not show the form used to assess the facility; it is the patient tracking form. Would the authors please include the form for facility assessment? Please also keep the patient tracking form as an appendix, as well as the line item form.

Response: We have revised S1 and S2 Appendices. S1 Appendix now includes the health facility assessment form, whereas S2 Appendix comprises both the patient tracking form and the line list.

4. 157: Please clarify for how much time there was a drug shortage. It would also help the reader to know how many days’ worth of medications each patient was given at every clinic visit and how often they were expected to return for follow-up.

Response: We thank Reviewer 1 for this comment. We now clarify that the drug shortage in one facility in Sud lasted only three days. No patient receiving treatment at this facility had treatment interruption as a result of this shortage (lines 163-165, page 7). We also clarify in the Methods the number of days of medications each patient is given at each clinic visit (lines 95-102, pages 4-5).

5. 173: Was the TB success rate determined only from PNLT data or from individual facility records, as well? When was the data accessed to determine treatment success? This would be prospective for most cases, since the target patients were identified within a few weeks after the hurricane and then followed to see if they were treated successfully. Was there a possibility that data entry into PNLT was delayed and results would have been different if the database had been accessed later?

Response: The treatment success rate was determined using data from the PNLT electronic database, not from individual facility records. The PNLT electronic database was accessed for treatment outcome analysis, after PNLT had validated its data using the facility TB case registers, symptoms registers, and lab registers. PNLT usually completes the validation process within six to nine months after the last patient initiates treatment. For this study, we accessed the outcomes data 12 months after the hurricane, when PNLT had already completed its validation, to ensure that outcome results would not have changed. 

6. 188-189: The difference could also have been due to treatment failure or not completing treatment, correct? Please clarify.

Response: We concur with Reviewer 1 that the difference could also have been due to adverse treatment outcomes (lost to follow up, died during treatment, not evaluated, and treatment failed) as described in the Methods section (lines 129-130, page 6). We have deleted this sentence to improve clarity.

7. 200 and Table 3: Please describe what was included in the multivariable analysis.

Response: We now describe in the Methods that the multivariable analysis included the following categorical variables: age, gender, commune of residence, smear status, category of patient (new vs. retreated patients), and HIV status (lines 134-138, page 6).

8. 221-222: Please explain the specific actions taken by TB providers. This will help readers consider how this could be replicated in similar settings.

Response: We thank Reviewer 1 for the suggestion. We now describe specific actions taken by providers in the Discussion section: (lines 250-254, page 13).

9. 234-235: If you choose to keep the comparison between assessed and unassessed facilities (the deeper dive into assessed facilities seems most important but this could be valuable as well), there is a need to further elucidate their differences. In other words, were assessed facilities larger than unassessed? Did they have more people from outside the commune of residence? What other aspects differed between the groups?

Response: Thank you for the suggestion to elucidate differences between assessed and unassessed facilities. Table 1 (page 9) now includes p-values, shown by asterisks, to show significant differences between characteristics of assessed and unassessed facilities. Assessed facilities had more patients residing in a different commune from the facility (12.3% vs. 4.7%), had a greater proportion of patients who were previously treated for TB, and a greater proportion of HIV-positive patients (11.4% vs 9.7%) or of patients who are of unknown HIV status (15.3% vs. 10.7%). These factors could have contributed to the lower treatment success observed in these facilities.

10. 258-260: The intent of this sentence is not clear. Would you kindly rewrite it to convey the information in a different way?

Response: This sentence has been rewritten to convey that the convenience sampling approach we used could have resulted in selection bias (lines 307-308, page 15).

11. In your conclusion, please provide specific recommendations (based on what was learned from this study) on how facilities and providers of TB care can be better prepared in case of future hurricanes or other disasters, acknowledging the limits of generalizability.

Response: Thank you for the suggestion. We now highlight in the conclusion (lines 316-319, pages 15-16) some of the key lessons learned that could help mitigate the impact of natural disasters on TB control in other resource-constrained settings. 

12. Beyond the outcome of treatment success, is there data available on how many of the patients experienced interruptions in treatment? If so, for how long were the interruptions? Were any at least 2 weeks in duration? Is so, was it the initiation or continuation phase? Did any develop drug resistance? These outcomes are important to report, if possible.

Response: We agree with Reviewer 1 that reporting treatment interruptions are important. We describe treatment interruptions and their durations (lines 168-170, page 8). Unfortunately, we do not have data on whether the interruptions were in the intensive or continuation phase as the data was collected in aggregate form and the PNLT electronic database does not capture interruptions or drug resistance.

13. In general, please scrutinize the grammar in detail as there are a few typos. In addition, please include the bracketed references before the period at the end of the sentence being referenced.

Response: Thank you for the suggestion. We have reviewed the grammar in detail to address typos. The bracketed references are now included before the period at the end of the sentence being referenced.

14. What are the possible reasons why patients with HIV or unknown HIV status had lower treatment success? Was this different before and after the hurricane or the same throughout?

Response: Thank you for these questions. We now include in the Discussion (lines 279-296, pages 14-15) a paragraph describing possible reasons why patients with HIV or unknown HIV status had lower treatment success. 

Comments from Reviewer 2

Major comments:

(1) better clarify the goals and objectives of the project and study and reformat the sections based on these goals and objectives,

(2) build up the discussion section further, and emphasize the significance and applicability of your findings to other similar settings outside of Haiti.

Response: We thank Reviewer 2 for the suggestions. We better clarify the goals and objectives of the project and study. We have reformatted the sections accordingly to reflect the changes. In addition, we have expanded the discussion section to comment on the role of HIV status on treatment success and to emphasize the significance and applicability of our findings to other resource-limited settings. 

Specific comments:

Introduction

• Lines 56-60 seem out of place and doesn’t really fit in with the flow of the introduction.

Response: We thank Reviewer 2 for pointing this out. We have removed lines 56-60 and the citation to the WHO guidelines. 

• Lines 61-63 The goal doesn’t seem to me as a goal but rather the activities you conducted in this important public heath surveillance project. It seems that the goal of this project was to evaluate the impact that hurricane Matthew had on TB services and outcomes in the three departments of Haiti. After you identify this goal, then you can delineate the specific objectives of this manuscript.

Response: We thank the Reviewer 2 for this suggestion. We have revised this section to incorporate the reviewer’s suggestion to state the goal and delineate the objectives of the study (lines 72-78, pages 3-4). 

• I would add a sentence emphasizing the importance of this study to the general reader, i.e. why are the findings important. I would focus on how many low resource settings are struck by both high TB prevalence and preponderance of devastating natural disasters and understanding how these phenomena are related can inform TB control activities.

Response: We have added sentences to emphasize the importance of the study to the general reader, describing how many low resource settings are struck by both high TB prevalence and vulnerability to devastating natural disasters (lines 65-71, page 3).

Methods

• Lines 73-75 I would be more specific on how this comparison is a representation of the impact of the hurricane on treatment outcomes. You could say something like “treatment started in quarter 1 would probably not be disrupted by the hurricane (unless the treatment was prolonged) and therefore represents a comparison group. While treatment started in quarters 2 and 3 represents the exposure group since treatment started in those quarters would have been disrupted since it would not have been completed until after the hurricane occurred.” I think this will make it clearer how you plan to compare treatment success before and after the hurricane.

Response: We thank Reviewer 2 for the suggestion to describe the exposure group in quarters 2 and 3 and the comparison groups in quarters 1 and 4. We have revised this section to make it clearer (lines 103-109, page 5).

• Lines 99-100 Please include citation for WHO definitions.

Response: We now include the citation for WHO definitions (lines 127 and 129, page 6).

• Lines 105-107 Did you use both chi square tests and odd ratios to determine association between risk factors and success. I would reword this sentence by saying you conducted bivariate analysis using odds ratios (and or chi square) with p-value<0.05.

• Lines 108-109 I am confused about the order of things in your analysis. It seems that from your results that you chose the appropriate epidemiologically relevant factors and conducted both bivariable and multivariable analysis on the same factors. If this is the case I would specifically say that.

Response: We have revised the “Data analysis” section to clarify that we used Pearson’s chi-squared testing to determine statistical differences between characteristics of patients in assessed facilities vs. those in unassessed facilities (lines 131-132, page 6). We also clarify that we conducted both bivariable and multivariable logistic regression to determine associations between epidemiologically relevant factors and treatment success (lines 134-138, page 6). 

Results

• Table 1 Should include chi-square testing with p-values to determine if there were statistical differences between those subjects assessed and those that weren’t assessed.

Response: We thank Reviewer 2 for this suggestion. Table 1 (page 9) now includes the p-values, with asterisks to denote statistical significance.

• Table 2 Should include chi-square testing with p-values to determine if there were statistical differences between treatment outcomes among the different sites. Also I would reword the title emphasizing that these are outcomes prior to the hurricane (i.e. Treatment initiated in quarters 1-3 before hurricane Matthew).

Response: Table 2 (page 10) now includes the p-value (denoted with an asterisk) to determine statistical significance. The title (lines 207-208) has been revised to emphasize that these are outcomes for patients receiving anti-TB treatment before hurricane Matthew.

• Lines 186-189 This assertion fits better in the discussion, specifically in the paragraph lines 229-233.

Response: We concur with Reviewer 2. This assertion has been rephrased and integrated in the Discussion section (lines 260-262, page 13).

• Table 3 I would eliminate the columns with p-values and denote the statistical significance of the ORs with asterisks. For example * for p-value<0.05 and ** for p-value<0.01.

Response: Thank you for the suggestion. We have eliminated the p-value columns from Table 3 (pages 11-12). We now denote statistical significance with asterisks.

Discussion

• Lines 225-228 Please include citation, whether it is a publication or based on anecdotal discussions with stakeholders

Response: We thank Reviewer 2 for pointing this out. We revise this as an observation during the facility assessments (lines 254-256, page 13).

• Lines 229-233 I know you mention in the introduction how the hurricane primarily stuck the areas that were assessed. But I would further describe how the geography of the hurricane, and perhaps other factors explain your findings, specifically that those in the assessed areas had a higher chance of treatment failure than those in areas that weren’t assessed. Perhaps you could use geospatial analysis to illustrate where the hurricane hit hardest and the percentage of treatment success in those areas.

Response: We appreciate the Reviewer’s suggestion to use geospatial analysis to determine whether treatment success in the areas hit hardest by the hurricane had lower treatment success rates in the assessed facilities. Treatment success for patients in the TB facility in the town where Hurricane Matthew made landfall (Les Anglais) was 97%. When we examined factors associated with lower treatment success rate after the hurricane in the assessed facilities, we found that those with moderate to severe damage were significantly more likely to experience lower treatment success rates after the hurricane (Odds ratio: 1.92; 95% confidence interval: 1.14-3.24; p = 0.015). Other factors that explain the lower treatment success in assessed facilities could be that these are referral facilities along major roads that received a high proportion of patients from other communes and had a higher proportion of patients with HIV or of unknown HIV status. 

• Lines 241-243, I would try to identify references or data from other settings that further explore the financial and travel-related restrictions placed on patients who live far from their healthcare facilities. This will add strength to your argument and perhaps improve the generalizability of your findings to other post-disaster settings.

Response: Thank you for the suggestion. We have added references from Haiti and other settings that support the argument that the cost of travel to health facilities can be a major impediment to anti-TB treatment (lines 272-273, page 14).

• Can you also comment on the role of HIV status on TB treatment success. You identify it as a statistically significant factor in the results but don’t mention it in the discussion.

Response: We now discuss the role of HIV status on TB treatment success in the discussion (lines 281-296, pages 14-15).

• Can you discuss further generalizability of your findings to other post disaster settings and possible limitation of applying your findings to other settings. An important take away readers will likely want to make from this study is how do these findings compare with those of other low resource, high burden Tb settings struck by natural disasters and how your findings could be used to inform mitigation efforts after future disasters.

Response: We thank Reviewer 2 for the insightful suggestions. We now include in the conclusion a statement about generalizability of our findings and key lessons that may be applicable to other resource-limited settings with a high TB burden that are prone to natural disasters (lines 313-321, pages 15-16).

---

## [Decision Letter · Decision Letter 1]

15 Feb 2021

Treatment success for patients with tuberculosis receiving care in areas severely affected by Hurricane Matthew – Haiti, 2016

PONE-D-20-06687R1

Dear Dr. Charles,

We’re pleased to inform you that your manuscript has been judged scientifically suitable for publication and will be formally accepted for publication once it meets all outstanding technical requirements.

Kind regards,

Anna Maria Mandalakas, MD

Guest Editor

PLOS ONE

Additional Editor Comments (optional):

Thank you for submitting your revised manuscript for consideration. The editors and the reviewers found the revised manuscript much improved and highly responsive to the original review comments. I am pleased to inform you that your revised manuscript has been accepted for publication in Plos One contingent upon completion of the few remaining reviewer comments shared below.

Reviewers' comments:

Reviewer's Responses to Questions

**Comments to the Author**

1. If the authors have adequately addressed your comments raised in a previous round of review and you feel that this manuscript is now acceptable for publication, you may indicate that here to bypass the “Comments to the Author” section, enter your conflict of interest statement in the “Confidential to Editor” section, and submit your "Accept" recommendation.

Reviewer #1: All comments have been addressed

Reviewer #2: All comments have been addressed

2. Is the manuscript technically sound, and do the data support the conclusions?

Reviewer #1: Yes

Reviewer #2: Yes

3. Has the statistical analysis been performed appropriately and rigorously? 

Reviewer #1: Yes

Reviewer #2: Yes

4. Have the authors made all data underlying the findings in their manuscript fully available?

Reviewer #1: Yes

Reviewer #2: Yes

5. Is the manuscript presented in an intelligible fashion and written in standard English?

Reviewer #1: Yes

Reviewer #2: Yes

6. Review Comments to the Author

Reviewer #1: Dear authors,

Congratulations on an excellent revision. I have two minor comments to share: 1) in Table 1, please be sure to place the correct symbols next to Accompagnateur and Previously treated; 2) please consistently use the "less than" (<) symbol to describe your p-values, instead of "less than or equal to."

Reviewer #2: The authors have done a nice job incorporating the reviewers' comments to make this a more rigorous and interesting manuscript. The coherence and structure of the manuscript is much improved. The relevance and generalizability to TB programs in other low resource settings is much more apparent.

There are still some sections where grammatical corrections should be made and arguments could be improved (see below) but otherwise I recommend this manuscript for publication.

Minor revisions:

Line 156-159: sentence is a little unclear. I had to read it a few times to understand it. I would add a word like "all" before the "20 in Sud" and "4 in Nippes" to emphasize that all facilities in those two departments suffered minimal to no damage while only a little over half the facilities in Grand'Anse had minimal or no damage.

lines 232 and 252: need to replace commas with "and"

lines 275-278: this paragraph seems like speculation. I recommend adding evidence to support this assertion. One possible suggestion is to analyze the sample of new TB patients following the hurricane and determine if there was an association between where these patients were seen and severity of damage on these facilities (i.e. was most of the drop in new patients seen at the facilities with severe damage). If so, you could combine the paragraphs 275-278 and 297-304 and make an assertion like "Persons with newly diagnosed TB and those already receiving care were both less likely to be seen at the facilities with severe damage..."

7. PLOS authors have the option to publish the peer review history of their article (what does this mean?). If published, this will include your full peer review and any attached files.

Reviewer #1: **Yes: **Karla Fredricks

Reviewer #2: No

---

## [Editor Report · Acceptance letter]

9 Mar 2021

PONE-D-20-06687R1 

Treatment success for patients with tuberculosis receiving care in areas severely affected by Hurricane Matthew – Haiti, 2016 

Dear Dr. Charles:

I'm pleased to inform you that your manuscript has been deemed suitable for publication in PLOS ONE. Congratulations! Your manuscript is now with our production department. 

Kind regards, 

on behalf of

Professor Anna Maria Mandalakas 

Guest Editor

PLOS ONE